# Variation of Soil Bacterial Communities in Forest Soil Contaminated with Chainsaw Lubricants

**DOI:** 10.3390/microorganisms12030508

**Published:** 2024-03-01

**Authors:** Ikhyun Kim, Manh Ha Nguyen, Sanggon Lee, Byoungkoo Choi, Keumchul Shin

**Affiliations:** 1Department of Forestry and Environmental Systems, Kangwon National University, Chuncheon 24341, Republic of Korea; kih9281@kangwon.ac.kr; 2Institute of Agriculture & Life Science, Gyeongsang National University, Jinju 52828, Republic of Korea; manhhafsiv@gmail.com; 3Forest Protection Research Center, Vietnamese Academy of Forest Sciences, Hanoi 11910, Vietnam; 4Department of Forest Environmental Resources, College of Agriculture and Life Sciences, Gyeongsang National University, Jinju 52828, Republic of Korea; tkdrhs170@naver.com; 5Division of Forest Science, Kangwon National University, Chuncheon 24341, Republic of Korea

**Keywords:** chainsaw, *Chthoniobacter*, lubricants, *Massilia*, soil bacterial community

## Abstract

Pollutants can exist in the soil for a long time and alter the bacterial community. Using lubricants to prevent the wear of chainsaw blades is necessary for thinning activities and wood harvesting. We investigated the influences of soil contamination with chainsaw lubricants on soil bacterial communities. Bio-oil, mineral oil, and recycled oil were scattered on each treatment to investigate variations in soil bacterial structure during treated periods using the Illumina MiSeq sequencing platform. The results obtained were 5943 ASVs, 5112 ASVs, and 6136 ASVs after treatment at one month, six months, and twelve months, respectively. There was a significant difference in Shannon and Simpson indices between treatments and controls. A total of 46 bacterial genera with an average relative abundance of more than 1.0% were detected in all soil samples. *Massilia* was the most common genus detected in control at one month, with an average relative abundance of 14.99%, while *Chthoniobacter* was the most abundant genus detected in bio-oil, mineral oil, and recycled oil treatments at one month, with an average relative abundance of 13.39%, 14.32%, and 10.47%, respectively. Among the three chainsaw lubricants, bio-oil and mineral oil had fewer impacts than recycled oil. The abundances of several functional bacteria groups in the bio-oil treatment were higher than in other treatments and controls. Our results indicated that different chainsaw lubricants and their time of application affected the soil bacterial community composition.

## 1. Introduction

Bacterial communities are essential to soil ecology, especially the forestry ecosystem [1,2]. They can degrade organic and inorganic compounds in the soil into soluble substances that trees can use for their development [3]. However, environmental conditions and human activities during production can impact soil bacterial communities [4,5,6]. The diversity of soil bacteria differed between forest types, soil depths, and slope positions [7]. Bacterial community structure also showed a significant difference among land use types, with a higher diversity in grassland soils than forest soils [1]. For croplands, soil bacterial communities were more influenced by winter than summer [8]. Changes in soil bacterial communities were recorded in contaminated soils, and pollutants mainly reduced the richness of soil bacterial structures [9]. Heavy metals have significantly impacted the taxonomic composition and diversity of bacterial communities in different soil types [10]. For example, contaminated soils with Cadmium (Cd) had changed the structure of soil bacteria in the croplands, with a decrease in genera such as *Lactococcus*, *Psychrobacter*, and *Brochothrix* [8]. The diversity of soil bacteria significantly decreased in the lead (Pb) and zinc (Zn) mining areas; however, some bacteria tended to increase and adapt to contaminated conditions, such as *Bradyrhizobium* spp., and they may be used as potential bioremediators in heavy metal-contaminated soils [11]. In addition, the application of pesticides is essential in preventing pests and plant diseases during agricultural production; however, pesticide residues have caused soil pollution and changed soil bacterial communities [12]. Several soil bacteria were absent after applying pesticide, but some other bacteria could survive as bioremediators in pesticide-contaminated soils [4,12].

In recent years, studies on the influence of persistent organic pollutants on soil bacterial communities have been conducted [13]. In which, changes in soil bacterial communities caused by polycyclic aromatic hydrocarbons (PAH) and total petroleum hydrocarbons (TPH) have been reported. PAH and TPH enter the soil through anthropological activities such as forest operations and harvesting agricultural products [9,14,15,16]. The dynamics of soil bacterial communities were related to PAH removal by phyto-microbial remediation, and the bacterial diversity of PAH-degraders can change over the course of treatment [9]. Hence, changes in bacterial communities can explain the bioremediation efficiency of PAH-contaminated soil [14]. Several potential bacteria that degraded TPH were enriched during soil contamination with diesel, such as *Paraburkholderia*, *Arthrobacter*, *Rhodanobacter*, and *Methylobacterium* [15]. In addition, *Paludibaculum*, *Marinobacter*, *Sphingomonas*, *Acidobacterium*, and *Rugosibacter* were the most dominant bacterial genera in diesel-contaminated soil that exhibited petroleum hydrocarbon biodegradability tolerance [16]. Using chain saws for thinning activities and wood harvesting is a common practice in the forest sector, and lubricants are necessary to prevent the wear of chain saw blades during this process. However, these lubricants can seep into the soil, causing contaminated soil and impacting soil respiration and seedling growth [17]. Assessing the influence of lubricants on changes in bacterial communities in forest soil is still limited and lacks information. Therefore, this study aimed to evaluate the effects of different chainsaw lubricants on soil bacterial communities. The experiment was established with three chainsaw oils, including biodegradable bar and chain oil (bio-oil), petroleum-based bar and chain oil (mineral oil), and recycled-petroleum-based bar and chain oil (recycled oil), and compared to the control. Soil sample collection and DNA extraction were carried out at different periods to compare the diversity and taxonomic structures among treatments through the Illumina MiSeq sequencing platform.

## 2. Materials and Methods

### 2.1. Experimental Design and Soil Sampling Collections

The experiment was established at a nursery field in the experimental forest of the Kangwon National University (37°46′46″ N, 127°49′42″ E) in Gangwon Province, Korea. The chainsaw lubricants were used for the experiment, including bio-oil (biodegradable bar and chain oil, a high-cost typical biodegradable bar-and-chain oil), mineral oil (petroleum-based bar and chain oil, a high-cost typical bar-and-chain oil), and recycled oil (recycled-petroleum-based bar and chain oil, a low-cost typical bar-and-chain oil). Three treatment plots and one control plot were designed, including three subplots (3 × 3 m) for each plot. Chainsaw oils were scattered on each subplot of treatments with a concentration of 100 mL/m^2^ using motor-operated spray. Soil samples were collected at three different periods after dispersing chainsaw lubricants for 1, 6, and 12 months. Soil samples were collected and mixed from three points at a 0–5 cm soil depth in each subplot. Approximately 300 g of soil was collected in each subplot and kept in brown glass bottles and sealed mouth bottles using Teflon tape to prevent photolysis and the air circulation of organic pollutants. The soil samples were maintained at 4 °C during transport to the laboratory and then stored at −20 °C until DNA extraction.

### 2.2. Soil DNA Extraction and Illumina MiSeq Sequencing

Soil bacterial DNA was extracted from soil samples (250 mg) using a DNeasy Power Soil Kit (Qiagen, Hilden, Germany), following the manufacturer’s instructions, and next-generation sequencing (NGS) was conducted. Quant-IT PicoGreen (Invitrogen, Waltham, CA, USA) was performed to measure the extracted genomic DNA concentration. Qualified samples were checked through quality control (QC) before 16S rRNA gene sequencing. The library construction proceeded according to the Illumina 16S metagenomics sequencing library preparation method [18] to amplify the bacterial 16S rRNA gene V3-V4 region. The first PCR was conducted with forward primers Bakt_341F (CCTACGGGNGGCWGCAG) and reverse primers Bakt_805R (GACTACHVGGGTATCTAATCC) [19]. AMPure beads (Agencour Bioscience, Beverly, MA, USA) were used to purify the first PCR products. Then, the first PCR products were used for secondary PCR to produce a sequencing library attached to a bar code using the Nextra XT index kit (Illumina, San Diego, CA, USA). The amplification products were measured in length and concentration by using TapeStation D1000 ScreenTape (Agilent Technologies, Waldbonn, Germany). The amplicon libraries were sequenced at Macrogen (Seoul, Republic of Korea) using an Illumina MiSeq^TM^ platform (Illumina, San Diego, CA, USA) according to the manufacturer’s protocol.

### 2.3. Sequencing, Processing, and Taxonomical Assignment

After completing sequencing, the raw sequences were sorted using index sequences and generated paired-end FASTQ files for each sample. Adapter sequences and used primers were removed using Cutadapt software (ver. 3.2) [20]. The error-corrected paired-end sequences were assembled into one sequence, and chimera sequences were removed using DADA2 to obtain the amplicon sequence variants (ASVs) [21]. Bacterial communities were comparatively analyzed using BLAST+ (version. 2.9.0) in the NCBI 16S microbial database and annotated with taxonomic information for the bacteria with the highest similarity [22].

### 2.4. Statistical Analysis

Species richness and diversity indices were calculated using the vegan package of R software (ver. 4.3.1), while principal coordinates analysis (PCoA) plots were constructed based on the UniFrac coefficient to indicate the differences in ASV compositions among treatments. The multiple data from different treatments were compared in a one-way ANOVA followed by Tukey’s HSD post hoc test to determine the significant differences between samples at a 5% probability level. Species diversity and the relative abundances of ASV proportions from different taxonomies were graphed using the ggplot2 package (ver. 3.4.3) in the R program [23].

## 3. Results

### 3.1. Effects of Chainsaw Lubricant Amendment on Bacterial Richness and Diversity

Analysis of the 16S rRNA gene sequence from high-throughput sequencing resulted in a total of 353,097, 472,209, and 354,116 read counts from soil samples (without chimeras) after treatment at one month, six months, and twelve months, respectively. These sequences were clustered into 5943 ASVs (one month), 5112 ASVs (six months), and 6136 ASVs (twelve months). All samples had good coverage values of more than 99%, which means the sequencing represents were sufficient to calculate soil bacterial diversity.

The number of ASVs obtained from different periods ranged from 540 to 1512 ASVs. Only the number of ASVs in the bio-oil treatment of 1-month samples had a significant difference (*p* < 0.05) compared to the control, while there was no significant difference between treatments and controls in other periods (Figure 1A). Similarly to observed ASVs, there was a significant difference (*p* < 0.05) in the average Chao1 richness between the bio-oil treatment and control of 1-month samples; in contrast, other comparisons in each period showed no significant difference (Figure 1B). From Figure 1C,D, it can be seen that chainsaw lubricant amendment affected Shannon and Simpson indices of bacterial communities in the soil samples. All three treatments had a higher level of bacterial diversity than controls at one month, with the highest level belonging to the bio-oil treatment; however, these differences in both Shannon and Simpson indices were insignificant among the treatments (Figure 1C,D). At six months, Shannon and Simpson indices in control were higher than those of the bio-oil and recycled oil treatments, and the mineral oil treatment had the highest indices. However, all treatments had no significant difference (Figure 1C,D). Bacterial diversity had a significant difference between the treatments at twelve months. Specifically, the Shannon and Simpson indices in control were the highest. At the same time, the lowest levels were indicated in the bio-oil treatment, and there was a significant difference (*p* < 0.05) between the control and bio-oil treatment (Figure 1C,D). The Simpson index in the mineral oil treatment also had a significant difference (*p* < 0.05) compared to the control, while the recycled oil treatment showed no significant difference compared to the control (Figure 1C,D). In general, bacterial diversity was affected by chainsaw lubricant amendments. Bio-oil and mineral oil had significant changes, while recycled oil had insignificant variations in the species richness and diversity index of the soil bacterial community.

### 3.2. Effects of Chainsaw Lubricant Amendment on Bacterial Community Composition

A total of 19 bacterial phyla were detected from all soil samples, of which the number of bacterial phyla obtained from bio-oil samples was also 19 phyla, followed by the control sample with 18 phyla. In comparison, mineral oil and recycled oil samples both had 17 phyla (Figure 2). The similarity of soil bacterial community composition between the controls and treatments was mainly Proteobacteria, Actinobacteria, Verrucomicrobia, Acidobacteria, Planctomycetes, Bacteroidetes, Chloroflexi, and Firmicutes; however, their relative abundances were different (Figure 2). The most dominant phyla were Proteobacteria, with relative abundances of 33.24–41.48%, 32.09–36.89%, and 31.17–35.60% at the one-month, six-month, and twelve-month samples, respectively. Actinobacteria accounted for the second dominant phyla in the six-month and twelve-month samples, with relative abundances of 28.67–31.64% and 13.48–20.36%, respectively, while this one-month sample belonged to Verrucomicrobia, with relative abundances of 13.12–16.80%. Elusimicrobia was only found in bio-oil samples, while Synergistetes occurred in both control and bio-oil samples with a very low relative abundance (Figure 2).

Changes in soil bacterial compositions were also shown at the class level. The number of bacterial classes with an average relative abundance of more than 1.0% was 13, 17, and 18 in the soil samples of one-month, six-month, and twelve-month periods, respectively (Figure 3). The relative abundance of bacterial classes was different among the treatments and periods. For example, Actinomycetia had the highest abundance in the six-month samples compared to other periods. In comparisons of one-month samples, Actinomycetia had the highest relative abundance (10.01%) in the control compared to bio-oil, mineral oil, and recycled oil. However, these bacteria were most dominant in the recycled oil treatments at six-month and twelve-month periods, with a relative abundance of 27.69% and 15.17%, respectively (Figure 3).

A total of 46 bacterial genera with an average relative abundance of more than 1.0% were detected in all soil samples (Figure 4). The number of genera ranged from 12 to 17 in one month samples, then tended to increase in the six month samples (24–27) and decrease in the 12 month samples (19–21). Bio-oil treatment had the highest number of genera, while recycled oil treatment had the lowest number of genera at one month. *Massilia* was the most common genus detected in control at one month, with an average relative abundance of 14.99%, while *Chthoniobacter* was the most abundant genus detected in bio-oil, mineral oil, and recycled oil, with an average relative abundance of 13.39%, 14.32%, and 10.47%, respectively. At six months, *Pseudarthrobacter* (5.57%) was the most abundant genus in control, while *Bradyrhizobium* was the most abundant genus detected in bio-oil, mineral oil, and recycled oil, with an average relative abundance of 5.29%, 5.90%, and 5.44%, respectively. For the twelve month period, *Paludibaculum* (4.44%) had the highest abundance in the control, *Chthoniobacter* was the most abundant genus in bio-oil and mineral oil with an average relative abundance of 7.38% and 5.74%, respectively, while *Acidobacterium* (4.22%) accounted for the highest relative abundance in the recycled oil treatment (Figure 4).

Principal Coordinate Analysis (PCoA) was conducted using the weighted UniFrac distances to visualize similarities in the soil bacterial community composition of different treatments at three sample collection periods. At one month, the contribution proportion of two axes, PC1 and PC2, accounted for 65.98% of the total variance of 12 variables, of which the rates of the PC1 and PC2 axes were 50.20% and 15.78%, respectively (Figure 5A). The division into groups did not differ much between the control and the treatments, and chainsaw lubricants had less effect on the soil bacterial community composition. At six months, the contribution rates of the PC1 and PC2 axes were 57.77% and 10.50%, respectively, accounting for 68.27% of the total variance of 12 variables (Figure 5B). Bio-oil and recycled oil treatments were divided into different groups with control, and these two treatments affected soil bacterial community composition. At twelve months, the contribution proportion of two axes, PC1 and PC2, accounted for 66.39% of the total variance of 12 variables, of which the rates of the PC1 and PC2 axes were 49.83% and 16.56%, respectively (Figure 5C). Control was divided into a separate group with treatments, and all three treatments affected soil bacterial community composition. These results showed that different chainsaw lubricants and their time of application affected the soil bacterial community composition.

## 4. Discussion

Soil bacteria are usually susceptible to habitat changes, even if the change is minimal [24,25,26]. Our study indicated that soil bacterial diversity indices differed between controls and treatments at various times (Figure 1). Changes in soil bacterial community structure by chainsaw lubricant amendment were also shown in the Principal Coordinate Analysis (PCoA) plot (Figure 5). The most significant difference was shown at twelve months when the diversity indexes in the treatments were lower than the control and the bacterial community composition in all treatments was different from the control. However, several beneficial bacteria in treatments at twelve months were more abundant than in the control (Table 1).

The diversity of soil bacteria can decrease in contaminated soils [10]. However, depending on pollutants, some functional bacteria can exist and develop to adapt to new conditions, and they play an essential role in the bioremediation process [27,28,29]. For example, *Mycolicibacterium* spp. can be found in PAH-contaminated soil and they can degrade PAHs [30,31]; *Acidibacter* and *Paludibaculum* were capable of dissimilatory ferric reduction [32,33]; *Aliidongia* and *Stella* produced poly-ꞵ-hydroxybutyrate, which can significantly contribute to the biodegradable plastics industry [34,35]; *Azospirillum* and *Vicinamibacter* increased drought and salinity tolerance and promoted plant growth [36,37,38]; *Fimbriiglobus* had a chitinolytic capability, while *Rhodoplanes* can produce hydrogen [39,40]; *Chthoniobacter* and *Bradyrhizobium* showed abilities of organic carbon degradation and nitrogen fixation, respectively [41,42].

**Table 1 microorganisms-12-00508-t001:** Relative abundance and functions of representative bacteria in soil samples of treatments at twelve months. The numbers in the same row with different letters showed significant differences among the treatments.

No.	Bacterial Genera	Functions	Relative Abundance (%) per Treatment			
			Control	Bio-Oil	Mineral Oil	Recycled Oil
1	*Acidibacter*	Dissimilatory reduction of ferric iron [32]	1.33 ^a^	1.33 ^a^	1.36 ^a^	1.15 ^a^
2	*Aliidongia*	Production of poly-β-hydroxybutyrate [34]	0.58 ^a^	1.26 ^b^	1.01 ^ab^	0.93 ^ab^
3	*Azospirillum*	Plant growth promotion [37]; Improving salinity tolerance of plant [36]	0.98 ^a^	1.71 ^b^	1.39 ^ab^	1.65 ^b^
4	*Bradyrhizobium*	Nitrogen fixation [42]	4.16 ^a^	4.18 ^a^	3.81 ^a^	3.64 ^a^
5	*Chthoniobacter*	Organic carbon degradation [41]	3.56 ^ab^	7.38 ^b^	5.74 ^ab^	2.94 ^a^
6	*Fimbriiglobus*	Chitinolytic capability [39]	0.54 ^a^	1.14 ^a^	1.06 ^a^	0.61 ^a^
7	*Mycolicibacterium*	PAH degradation [43,44]	1.40 ^a^	1.94 ^a^	1.99 ^a^	1.90 ^a^
8	*Paludibaculum*	Dissimilatory reduction of ferric iron [33]	4.24 ^a^	3.78 ^a^	4.29 ^a^	3.50 ^a^
9	*Rhodoplanes*	Hydrogen production [40]	3.28 ^a^	5.51 ^a^	4.90 ^a^	3.83 ^a^
10	*Stella*	Production of poly-β-hydroxybutyrate [35]	0.78 ^a^	1.31 ^a^	1.06 ^a^	1.01 ^a^
11	*Vicinamibacter*	Improving drought tolerance of plant [38]	0.99 ^a^	1.11 ^a^	1.34 ^a^	1.02 ^a^

Among the eleven functional bacteria in Table 1, their relative abundance in the bio-oil and mineral oil treatments was higher than in the recycled oil treatment. Three bacterial genera, *Aliidongia*, *Azospirillum*, and *Chthoniobacter*, had significant differences in relative abundance among the treatments and controls. *Chthoniobacter* was most commonly found in the bio-oil treatment with a relative abundance of 7.38%, followed by the mineral oil treatment with a relative abundance of 5.74%; the control had a lower percentage than the bio-oil and mineral oil treatments with a relative abundance of 3.56%, and the recycled oil treatment had the lowest proportion of 2.94% (Table 1). *Chthoniobacter flavus* belongs to the Spartobateria class within the Verrucomicrobia phylum and was found in ryegrass and clover pasture [41,45]. This species can grow on different saccharides and be considered a biotic agent that recycles carbon from plant biomass [41,45]. *Chthoniobacter flavus* was the most abundant in municipal solid waste from Nagpur, Maharashtra, India, and this bacterium can degrade organic pollutants through the biostimulation process [46]. *Chthoniobacter* was one of the dominant genera in soil from the indigenous and commercial forests in South Africa [47]. The abundance of *Chthoniobacter* in organic cropping systems was higher than that in conventional cropping systems [48]. Using biochar to control bacterial wilt disease caused by *Ralstonia solanacearum* also significantly increased the abundance of *Chthoniobacter* in crop soil in Southern China [49].

The relative abundance of *Azospirillum* was detected with higher percentages in all treatments than in control, and bio-oil had the highest proportion of 1.71%, followed by recycled oil, mineral oil, and control, with proportions of 1.65%, 1.39%, and 0.98%, respectively (Table 1). The endophytic bacterium *Azospirillum* sp. isolated from rice plants can enhance rice growth and resistance against rice blast disease [50,51,52]. *Azospirillum* biofertilizer can be used to remove copper and chromium in the aqueous medium [53], and *Azospirillum* spp. has also been recorded to increase wheat’s water stress tolerance and lettuce’s saline tolerance [36,54]. *Azospirillum Brasiliense*, when combined with bentonite, can improve soil quality and maximize wheat yields up to 42% [55]. *A. brasiliense* was also a potential agent to promote the productivity of sugarcane (*Saccharum* spp.), and it can offer a promise for bioenergy strategy by enhancing the valuable chain of sugarcane productions [56]. In addition, *A. brasiliense*, isolated from Vietnamese wet rice, can produce indole-3-acetic acid [57]. *Azospirillum argentinense* inoculation of maize seeds made maize plants grow faster, and the yield of maize (*Zea mays*) was also enhanced after treatment [58].

*Aliidongia* genus was also detected as the most abundant in the bio-oil treatment, with a relative abundance of 1.26%, significantly higher than the control with a rate of 0.58%. The mineral oil and recycled oil treatments had relative abundances of 1.01% and 0.93%, respectively; however, there was no significant difference between these two treatments and the control (Table 1). *Aliidongia dinghuensis*, a gram-negative bacterium, was found in soil samples of *Pinus massiniana* plantations; this bacterium produced poly-β-hydroxybutyrate and can be applied to the biodegradable plastics industry [34]. Discovering soil bacterial communities in the bulk soil and rhizosphere soil of *Carya cathayyensis* influenced by root rot disease showed that the relative abundance of the *Aliidongia* genus was higher in bulk soil than in rhizosphere soil [59]. The abundance of *Aliidongia* in boreal forest soil was also significantly affected by harvesting and forest floor removal [60].

In general, bio-oil, mineral oil, and recycled oil have changed the structure of the soil bacterial community. Of which, bio-oil had higher relative abundances in several functional bacteria than mineral oil and recycled oil. In addition, contaminated soil can facilitate faster recovery using bio-oil than mineral oil and recycled oil [17]. Therefore, using bio-oil may be the optimal solution and have the most negligible impact on the microbial community in the soil. To develop biological treatments for soil contamination, the isolation of functional bacteria and assessment of their potential applications should be carried out in further studies.

## 5. Conclusions

The results of this study suggested that chainsaw lubricants can change soil bacterial community structure over time. In one month, the diversity of soil bacteria in treatments was higher than in the control. However, soil bacteria’s species richness and diversity in treatments decreased gradually at six months and were lower than in the control at twelve months. Different chainsaw lubricants and their time of application affected the soil bacterial community composition. Among the three chainsaw lubricants, bio-oil and mineral oil had fewer impacts than recycled oil. The abundances of several functional bacteria groups in the bio-oil treatment were higher than in other treatments and controls. After twelve months of treatment, *Aliidongia*, *Azospirillum*, and *Chthoniobacter* had significantly different abundances among treatments, and these bacteria can significantly contribute to the removal process of pollutants from contaminated soils. The highest abundances of *Aliidongia*, *Azospirillum*, and *Chthoniobacter* were indicated in bio-oil treatment, with relative abundances of 1.26%, 1.71%, and 7.38%, respectively. Hence, bio-oil may be recommended for further forestry operations to minimize negative influences on soil microbial communities.

## Figures and Tables

**Figure 1 microorganisms-12-00508-f001:**
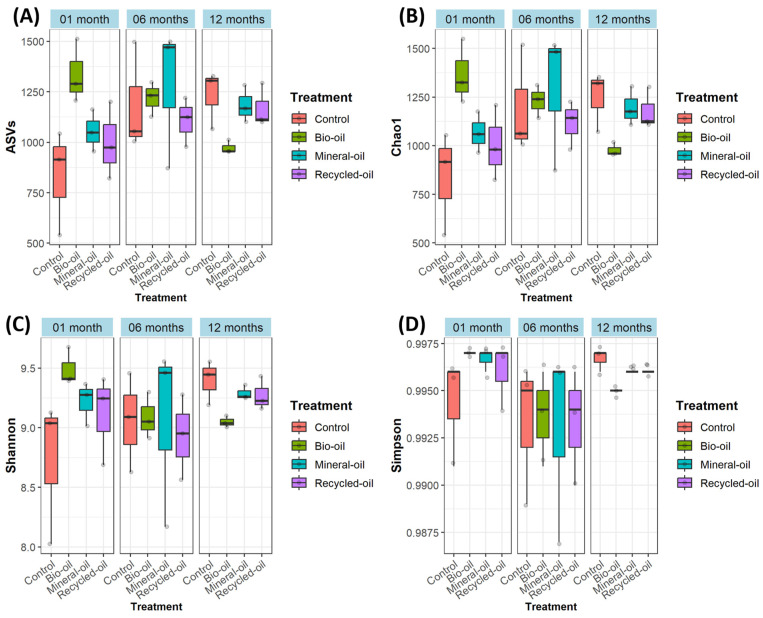
The species richness and diversity indices of soil bacteria among treatments at different collected sample times. The number of amplicon sequence variants (ASVs) (**A**); Chao1 richness (**B**); Shannon index (**C**); Simpson index (**D**).

**Figure 2 microorganisms-12-00508-f002:**
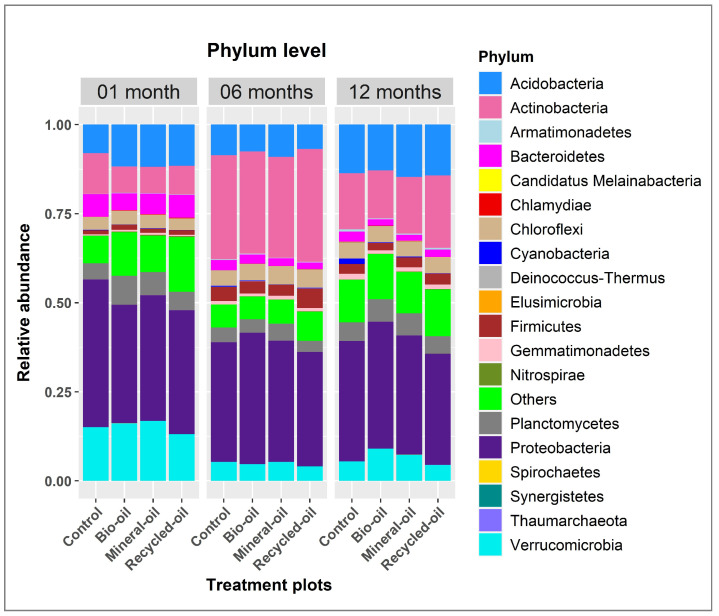
Relative abundance of soil bacteria at the phylum level in different soil samples treated with bio-oil, mineral oil, recycled oil, and control during contaminated chainsaw lubricant periods.

**Figure 3 microorganisms-12-00508-f003:**
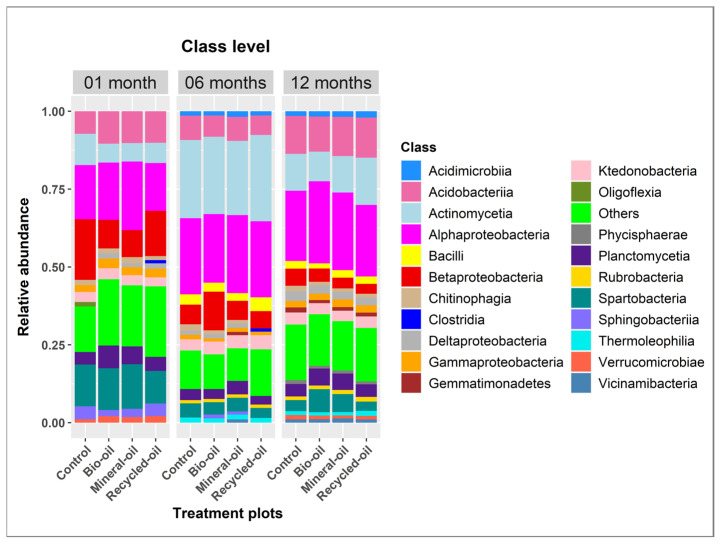
Relative abundance of soil bacteria at the class level in different soil samples treated with bio-oil, mineral oil, recycled oil, and control during contaminated chainsaw lubricant periods. The bacteria with a relative abundance below 1% were grouped into others.

**Figure 4 microorganisms-12-00508-f004:**
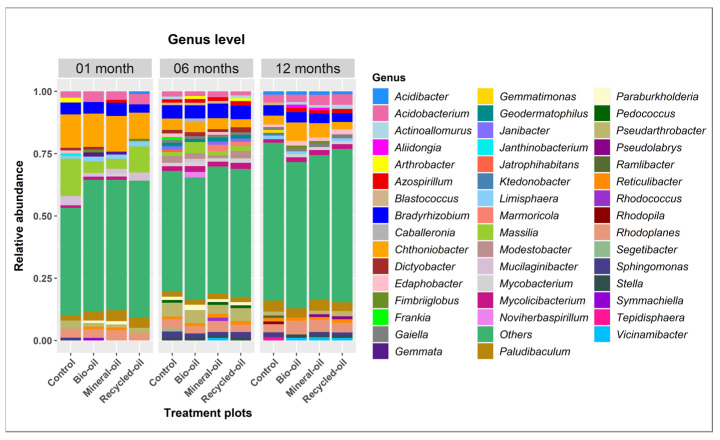
Relative abundance of soil bacteria at the genus level in different soil samples treated with bio-oil, mineral oil, recycled oil, and control during contaminated chainsaw lubricant periods. The bacteria with a relative abundance below 1% were grouped into others.

**Figure 5 microorganisms-12-00508-f005:**
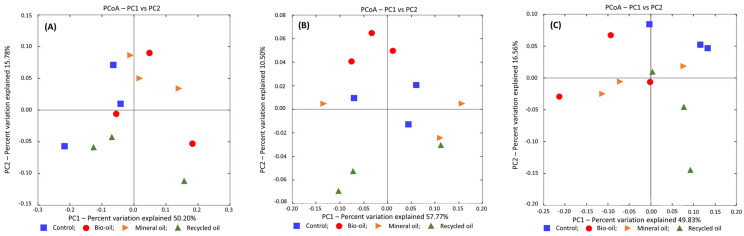
Principal Coordinate Analysis (PCoA) based on the weighted UniFran distances for bacterial communities from treatments and control at different periods, (**A**): 1 month; (**B**): 6 months; (**C**): 12 months.

## Data Availability

The data presented in this study are available on request from the corresponding authors. The data are not publicly available due to institutional policy.

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
