# Peer review of "Variation of Soil Bacterial Communities in Forest Soil Contaminated with Chainsaw Lubricants"

_microorganisms, 2024, doi:10.3390/microorganisms12030508_

Round 1

Reviewer 1 Report

Comments and Suggestions for Authors

Manuscript entitled “Variation of Soil Bacterial Communities in Forest Soil Contaminated with Chainsaw Lubricants” submitted by Ikhyun Kim, Manh Ha Nguyen, Sanggon Lee, Byoungkoo Choi and Keumchul Shin, cannot be considered for publication in Microorganisms Journal, in this form.

Here is a list of my specific comments:

  1. Page 1, Abstract: This section should be rewritten. Include in this section the most important results and findings to highlight the importance of this study.
  2. Page 1, line 21: “We assume soil bacterial communities…”. This is well known. This paragraph should be reworded.
  3. Page 1, 1. Introduction: This section must be reorganized. The most important aspects related to this topic should be presented to provide a clear description of the state of art in this field.
  4. Page 1, line 47: “For example, contaminated soils…”. These examples should be deleted, because are irrelevant for this study.
  5. Page 2, 2. Materials and Methods: The analysis of the soil samples should be also done. Therefore, include in this section a sub-section and describe this methodology.
  6. Page 2, line 79: “…including bio-oil…”. The main characteristics of these oils must be presented here.
  7. Page 3, 3. Results: Provide a clear and logical presentation of the experimental results. Move all important observations in the next section.
  8. Page 7, 4. Discussion: In this section, all discussions should be supported by the experimental results. If not, they are irrelevant. Therefore, delete general observations and pay attention on discussing the results presented in the previous section.
  9. Page 9, 5. Conclusions: Include in this section the most important experimental results and findings to highlight the importance of this study.

Author Response

Responses to Reviewer 1

We are grateful for the constructive evaluations by the reviewer and for the helpful comments. In the following, changes between the previous and the new submission are listed.

General reviewer comment:

Manuscript entitled “Variation of Soil Bacterial Communities in Forest Soil Contaminated with Chainsaw Lubricants” submitted by Ikhyun Kim, Manh Ha Nguyen, Sanggon Lee, Byoungkoo Choi and Keumchul Shin, cannot be considered for publication in Microorganisms Journal, in this form.

Response: We have improved our manuscript following your detailed comments.

Detailed reviewer comment #1

Page 1, Abstract: This section should be rewritten. Include in this section the most important results and findings to highlight the importance of this study.

Response: We added and changed some text with a blue color in the revised manuscript to show the most important results and findings of our study.

Detailed reviewer comment #2

Page 1, line 21: “We assume soil bacterial communities…”. This is well known. This paragraph should be reworded.

Response: We added and changed some text with a blue color in the revised manuscript.

Detailed reviewer comment #3

Page 1, 1. Introduction: This section must be reorganized. The most important aspects related to this topic should be presented to provide a clear description of the state of art in this field.

Response: We added some information to provide a clear description of the state of art in this field with a blue color in the revised manuscript.

Detailed reviewer comment #4

Page 1, line 47: “For example, contaminated soils…”. These examples should be deleted, because are irrelevant for this study.

Response: We want to keep this information to provide a minor overview of bacterial communities in contaminated soil.

No changes were made.

Detailed reviewer comment #5

Page 2, 2. Materials and Methods: The analysis of the soil samples should be also done. Therefore, include in this section a sub-section and describe this methodology.

Response: We used soil samples for DNA extraction so we added some text to describe the detailed method of soil sample collection with a blue color in the revised manuscript.

Detailed reviewer comment #6

Page 2, line 79: “…including bio-oil…”. The main characteristics of these oils must be presented here

Response: We added some text to describe main characteristics of oils with a blue color in the revised manuscript.

Detailed reviewer comment #7

Page 3, 3. Results: Provide a clear and logical presentation of the experimental results. Move all important observations in the next section.

Response: We want to keep the results structure because we show logical results of alpha and beta diversity of the bacterial community.

No changes were made.

Detailed reviewer comment #8

Page 7, 4. Discussion: In this section, all discussions should be supported by the experimental results. If not, they are irrelevant. Therefore, delete general observations and pay attention on discussing the results presented in the previous section.

Response: We discussed data from Figures 1 and 5 and Table 1. The data in Table 1 was summarized from Figure 4 and referred to published papers. Therefore, in our opinion, they are relevant to the results.

No changes were made.

Detailed reviewer comment #9

Page 9, 5. Conclusions: Include in this section the most important experimental results and findings to highlight the importance of this study.

Response: We added more text to show more detailed conclusions with a blue color in the revised manuscript.

Reviewer 2 Report

Comments and Suggestions for Authors

The manuscript contains valuable information on changes in the soil prokaryotic community under the influence of different types of lubricants application. The results obtained have both theoretical and practical significance for predicting changes in the soil microbiome as a result of human management activities. The article can be published after a minor revision in accordance with the comments in the attached file. 

Author Response

Responses to Reviewer 2

We are grateful for the constructive evaluations by the reviewer and for the helpful comments. In the following, changes between the previous and the new submission are listed.

General reviewer comment:

The manuscript contains valuable information on changes in the soil prokaryotic community under the influence of different types of lubricants application. The results obtained have both theoretical and practical significance for predicting changes in the soil microbiome as a result of human management activities. The article can be published after a minor revision in accordance with the comments in the attached file. 

Detailed reviewer comment #1

Line 28: not in "oil" itself, you mean in "bio-oil, mineral oil etc" treatments?

Response: Yes, we added “treatments” in the text with a blue color in the revised manuscript.

Detailed reviewer comment #2

Line 30: several functional groups?

Response: Yes, we added “groups” in the text with a blue color in the revised manuscript.

Detailed reviewer comment #3

Line 85: 1, 6 will look better

Response: We changed the text with a blue color in the revised manuscript.

Detailed reviewer comment #4

Line 85: 300 g is a mixed sample from the whole plot? Please describe in detail how the samples were collected.

Response: We added more text to describe in detail how the samples were collected with a blue color in the revised manuscript.

Detailed reviewer comment #5

Line 90: What was the amount/concentration of isolated dna

Response: the amount/concentration of isolated dna was measured by Macrogen in Seoul before sequencing and all the concentrations of dna were passed for the quality check for the sequencing.

Detailed reviewer comment #6

Did you deposite your raw sequences in commonly accessible database? NCBI Sequence Read Archive, etc..?

Response: No, we did not deposit the raw sequences from the metagenomics analysis into NCBI. They are huge data sets from each sample, which are usually not deposited for these types of manuscript work.

Detailed reviewer comment #7

Line 231: Please improve the resolution of the figure.

Response: We changed figures with a higher resolution in the revised manuscript.

Detailed reviewer comment #8

Line 269: remove italics

Response: In the revised manuscript, we changed the word to a regular type with a blue color.

Reviewer 3 Report

Comments and Suggestions for Authors

The article is devoted to a very interesting and relevant topic: studying the influence of oil from a chainsaw after cutting down trees. The microbial community reacts very sensitively to the presence of foreign organic matter in the soil.

The article is written in scientific language and has interesting results.

I recommend the article for publication.

Author Response

Responses to Reviewer 3

General reviewer comment:

The article is devoted to a very interesting and relevant topic: studying the influence of oil from a chainsaw after cutting down trees. The microbial community reacts very sensitively to the presence of foreign organic matter in the soil.

The article is written in scientific language and has interesting results.

I recommend the article for publication.

Response: Thank you very much.

We have changed some texts to correct grammatical errors and for other reviewers’ recommendations.

Reviewer 4 Report

Comments and Suggestions for Authors

Article title:

“Variation of Soil Bacterial Communities in Forest Soil Contaminated with Chainsaw Lubricants".

Three types of chainsaw oils were used in the experiment: recycled petroleum-based bar and chain oil (recycled oil), petroleum-based bar and chain oil (mineral oil), and biodegradable bar and chain oil (bio-oil). These oils were compared to a control group. Using the Illumina MiSeq sequencing platform, soil sample collection and DNA extraction were done at various times to assess taxonomic structures and diversity between treatments.

The work done is certainly of international interest and the format applied is certainly suitable for a review. This review dealt with the topic differently and attractively, and the titles are related to each other. The work is original, of particular interest, and can certainly stimulate research on this topic. The conclusion summarizes the aims of the work and prospects.

Author Response

Responses to Reviewer 4

General reviewer comment:

“Variation of Soil Bacterial Communities in Forest Soil Contaminated with Chainsaw Lubricants".

Three types of chainsaw oils were used in the experiment: recycled petroleum-based bar and chain oil (recycled oil), petroleum-based bar and chain oil (mineral oil), and biodegradable bar and chain oil (bio-oil). These oils were compared to a control group. Using the Illumina MiSeq sequencing platform, soil sample collection and DNA extraction were done at various times to assess taxonomic structures and diversity between treatments.

The work done is certainly of international interest and the format applied is certainly suitable for a review. This review dealt with the topic differently and attractively, and the titles are related to each other. The work is original, of particular interest, and can certainly stimulate research on this topic. The conclusion summarizes the aims of the work and prospects.

Response: Thank you very much.

We have changed some texts to correct grammatical errors and for other reviewers’ recommendations.

Round 2

Reviewer 1 Report

Comments and Suggestions for Authors

All my previous remarks and comments have been considered in this new version of the manuscript. In my opinion, the revised manuscript meets the criteria and can be published as original paper in Microorganisms Journal.